# Association between Symptoms of Depression and Generalised Anxiety Disorder Evaluated through PHQ-9 and GAD-7 and Anti-Obesity Treatment in Polish Adult Women

**DOI:** 10.3390/nu16152438

**Published:** 2024-07-26

**Authors:** Tomasz Witaszek, Karolina Kłoda, Agnieszka Mastalerz-Migas, Mateusz Babicki

**Affiliations:** 1Tomasz Witaszek-Gabinet Leczenia Otyłości, ul. Józefińska 33/8, 30-529 Kraków, Poland; 2MEDFIT Karolina Kłoda, ul. Narutowicza 13E/11, 70-240 Szczecin, Poland; wikarla@gazeta.pl; 3Department of Family Medicine, Faculty of Medicine, Wroclaw Medical University, 50-367 Wrocław, Poland; agnieszka.migas@gmail.com (A.M.-M.); ma.babicki@gmail.com (M.B.)

**Keywords:** obesity, anxiety, depression, women health, BMI

## Abstract

Obesity impacts mental health greatly. Psychological factors may influence the effectiveness of its treatment. This study aimed to compare symptoms of generalised anxiety disorder and depression among adult women across different weight categories. The study sample comprised 1105 adult women. The computer-assisted web interview (CAWI) utilising the seven-item Generalised Anxiety Disorders Scale (GAD-7) and the nine-item Patient Health Questionnaire (PHQ-9) was used. Both GAD-7 and PHQ-9 scores correlated positively with BMI (r = 0.121, *p* < 0.001 and r = 0.173, *p* < 0.001, respectively) and negatively with age (r = −0.106, *p* < 0.001 and r = −0.103, *p* < 0.001, respectively). Patients undergoing treatment with semaglutide scored lower for both anxiety symptoms (8.71 ± 6.16, *p* = 0.013) and depression symptoms (9.76 ± 6.37, *p* = 0.013). Women who underwent bariatric surgery screened less frequently for anxiety (8.03 ± 6.27, *p* = 0.002) but not for depression. An interdisciplinary approach involving mental health professionals within the therapeutic team can comprehensively address factors contributing to obesity development and treatment outcomes. Further investigation of semaglutide’s use is needed due to the promising evidence suggesting a positive effect on decreasing the severity of depression and anxiety symptoms to assess the direct or indirect character of this influence.

## 1. Introduction

Obesity, a multifactorial and complex disease, has reached pandemic proportions. According to the WHO report from 2022, one in eight people in the world lives with obesity. Its prevalence varies by region, from 31% in the South-East Asia region and the African region to 67% in the region of the Americas [1]. As the number of individuals with obesity increases, so does the prevalence of the associated consequences of excess body weight. In particular, obesity has an important contribution to the global incidence of cardiovascular disease, type 2 diabetes mellitus, cancer, osteoarthritis, work disability, and sleep apnoea. It has a more pronounced impact on morbidity than on mortality [2]. Its physical comorbidity burden is well-researched [3,4]. However, obesity also has a significant impact on mental health. In recent years, there has been an increasing focus on the relationship between body weight and mental health disorders, driven by their rising prevalence, socioeconomic factors, and heightened public awareness (including the influence of social media, excessive focus on body image, body dysmorphophobia, and misinterpretation of healthy weight standards) [5,6,7,8,9].

A meta-analysis conducted in 2010 by Luppino et al. [6] reported a bidirectional association between depression and obesity. Over time, individuals living with obesity had a 55% increased risk of developing depression, while individuals with depression had a 58% increased risk of obesity. Importantly, the association between depression and obesity surpassed that of depression and overweight, reflecting a dose-response gradient. In a different meta-analysis from 2017 by Jung et al. [7], obesity was also found to increase the risk of depression. Individuals with more severe obesity (BMI ≥ 40 kg/m^2^) exhibited a stronger association, once again indicative of a dose-response pattern. Studies further suggest that the association between obesity and depression is more pronounced in women than in men [6,7,8]. It has been hypothesised that the current body ideal, emphasising thinness, affects women more than men and causes more psychological distress, which can lead to depression. In contrast, men who are overweight showed a significantly decreased risk of depression [7]. This phenomenon has been explained in prior research through the “jolly fat hypothesis”, which posits a negative association between high body weight and depressive symptoms in men [10]. There were significantly higher rates of overweight and obesity in transgender population, with the results being more pronounced in those assigned female at birth [11]. The co-occurrence of depression and obesity can also complicate the recognition and treatment of its complications, such as sleep apnoea [12]. The observed variations in results across different continents can suggest that the relationship between body size and depression may be influenced by the cultural conception of the ideal body weight and related social pressures on the relationship between body size and depression. In cultures where a larger body size is considered the norm, obesity might be more socially acceptable, leading to lower levels of body dissatisfaction, reduced mental stress, and becoming a buffer against weight discrimination [7,13]. 

The association between body weight and anxiety disorder is less studied, which can be due to its less direct and measurable impacts on mortality and morbidity. However, a recent meta-analysis revealed that individuals with obesity and overweight have higher levels of anxiety compared with those without obesity and that this relationship is stronger in women than in men [14]. Obesity increases the odds of an anxiety disorder or anxiety symptoms (such as dread or unease) by 30% and can be predictive of their chronic course [15]. The relationship between excess body weight and anxiety can be the result of one of the maladaptive eating patterns called emotional eating (food intake as an emotional response to negative excitement). This behaviour has been established as a contributing factor to weight gain [16]. Anxiety also has a significant positive correlation with metabolic syndrome [17]. Interestingly, OCD is associated with significantly lower rates of obesity and overweight, but this relationship was not found when comorbid depression was present [18]. 

Considering the broader context of obesity treatment, it becomes evident that psychological factors play an important role. There is a variety of treatments that can be offered to patients with obesity, ranging from lifestyle modifications and dietary interventions to pharmacotherapy and surgical options. In recent years, non-invasive methods of treating obesity have become increasingly effective, especially due to rapid advancements in the development of new incretin-based medications [19]. People living with obesity seeking treatment have more psychopathologies, such as anxiety, depression, eating pattern problems, and lower levels of self-esteem than normal-weight controls [20]. These psychological factors may have an influence on the effectiveness of obesity treatment, whether surgical or non-surgical [21]. A better understanding of how mental disorders interact with various treatment methods is needed. 

The aim of this study was to compare adult women across different weight categories —normal weight, overweight, and obesity in regard to symptoms of generalised anxiety disorder and depression evaluated through PHQ-9 and GAD-7 questionnaires, respectively. In the case of women living with overweight and obesity, we explored the associations between scores on validated screening tools, the use of anti-obesity medications, and the status of past bariatric surgery. We hypothesised that there would be an association between body weight and scores on these questionnaires and that the significance of this association would increase with an increasing BMI. We anticipated that the use of anti-obesity medication and the bariatric surgery status would influence the scores of GAD-7 and PHQ-9.

## 2. Materials and Methods

### 2.1. Participants and Recruitment 

The data for this study were collected through a computer-assisted web interviewing (CAWI) survey using a proprietary questionnaire created on Google Forms that included standardised psychometric tools: the PHQ-9 and GAD-7 scales. The survey took place between 4 September 2023 and 19 October 2023 and was distributed via various social media platforms, such as Facebook and Instagram. The target groups included members of support groups focusing on healthy lifestyle advice, bariatric surgery, and the use of anti-obesity medications. Participants were fully informed about the study’s purpose and methodology and provided informed consent before participating. They had the option to withdraw at any time. Inclusion criteria included being female, aged 18 or older, residing in Poland, and having internet access. The exclusion criteria were being male, not giving consent, and providing incompetent responses in the questionnaire. The participants were divided into six groups based on their calculated body mass index (BMI): underweight (BMI below 18.5), normal weight (BMI 18.5 to 24.9), overweight (BMI 25.0 to 29.9), Obesity I (BMI 30.0 to 34.9), Obesity II (BMI 35.0 to 39.9), and Obesity III (BMI above 40) [22]. The study adhered to the principles of the Declaration of Helsinki and received approval from the Bioethics Committee of the Wroclaw Medical University, Poland (approval number: 349/2023N). The questionnaire began by gathering socio-demographic information such as age, gender, height, current weight, and highest recorded weight. Participants were then asked about any chronic conditions, their use of anti-obesity medications, and their bariatric treatment status. Those who answered positively were further queried about specific comorbid conditions, anti-obesity medications, and types of bariatric procedures they underwent. The second part of the survey included standardised psychometric tools.

### 2.2. PHQ-9 Scale 

The Patient Health Questionnaire (PHQ-9) is a widely used tool for diagnosing and assessing the severity of depression symptoms. It asks individuals to rate on a four-point scale ranging from “not at all” to “most days” how frequently they’ve experienced specific depression symptoms over the past two weeks. Researchers have confirmed the validity and reliability of the PHQ-9 [23]. A validated Polish version of the questionnaire was utilised, with a cut-off value of 12 points determined [24]. The tool’s reliability, measured using Cronbach’s alpha coefficient, was found to be 0.872.

### 2.3. GAD-7 Scale

The seven-item Generalised Anxiety Disorder Scale (GAD-7) was developed as a screening tool designed to identify generalised anxiety disorder (GAD), particularly in primary care settings [25]. It assesses the frequency of experiencing seven distinct symptoms of anxiety over the past two weeks. Response options include “not at all”, “several days”, “more than half the days”, and “nearly daily”, scored as 0, 1, 2, and 3, respectively [25]. The Polish translation of the GAD-7, provided by the MAPI Research Institute, was used [26]. Internal consistency, measured using Cronbach’s alpha, was found to be high at 0.924, indicating strong reliability. It is crucial to ensure the psychometric robustness of these tools, as demonstrated in various contexts [27]. For a better understanding of the research methodology, the English version of the survey was attached in Appendix A (the English version of the study questionnaire).

### 2.4. Statistical Analysis 

The study involved variables with both qualitative and quantitative attributes. The normality of the distribution was assessed using the Shapiro–Wilk test. Qualitative variable comparisons were performed using the chi-squared test, while for quantitative variables, non-parametric tests such as the Kruskal–Wallis H Test or the Mann–Whitney U test were employed. The level of correlation between variables was evaluated using the Spearman’s correlation coefficient (rS). A significance level of 0.05 was set. Statistical analyses were conducted using Statistica 13.0 (TIBCO Software, Palo Alto, CA, USA). 

## 3. Results

### 3.1. Characteristics of the Study Group

We analysed responses from 1105 female participants, with a mean age of 38.89 ± 9.0 years and a mean BMI of 33.32 ± 6.67 kg/m^2^. Within the study population, 29.6% had chronic diseases, with obesity being the most prevalent (44.8%, n = 496). A total of 19.1% (n = 211) of the respondents had depression, while 13.8% (n = 152) had an anxiety disorder diagnosis. Prescription anti-obesity medication was actively used by 44.2% of women, while 11.9% of women (n = 131) underwent bariatric surgery. A comprehensive breakdown of the study group’s characteristics is provided in Table 1.

Among all participants, 46.4% (n = 513) screened positively for anxiety, and 50.3% (n = 556) screened positively for depression. Results of the Generalised Anxiety Disorder and Patient Health Questionnaire-9 and their interpretation are presented in Table 2. 

### 3.2. Comparison of Generalised Anxiety Disorder and Depression Symptoms Evaluated through GAD-7 and PHQ-9 among Adult Women with Overweight, Obesity, and Normal Weight

Both GAD-7 and PHQ-9 scores correlated positively with BMI (r = 0.121, *p* < 0.001 and r = 0.173, *p* < 0.001, respectively) and negatively with age (r = −0.106, *p* < 0.001 and r = −0.103, *p* < 0.001, respectively). Correlations between PHQ-9 and GAD-7 scores with age and BMI are presented in Table 3.

When the study group was divided into five subgroups based on BMI (normal weight, overweight, and obesity grades I to III), a significant relationship between obesity grade and both GAD-7 and PHQ-9 scores (*p* < 0.001) was observed. As BMI increased, so did the scores and the number of participants meeting the cut-off threshold. Patients undergoing treatment with semaglutide exhibited lower scores for both anxiety symptoms (8.71 ± 6.16, *p* = 0.013) and depression symptoms (9.76 ± 6.37, *p* = 0.013). However, no significant relationships were found between other anti-obesity medications. Women who underwent bariatric surgery screened less frequently for anxiety (8.03 ± 6.27, *p* = 0.002) but not for depression. A detailed overview of the results of GAD-7 and PHQ-9 is presented in Table 4 and Table 5, distinguishing between different BMI subgroups.

## 4. Discussion

This study aimed to compare adult women of varying weight categories in terms of symptoms of generalised anxiety disorder and depression assessed via the PHQ-9 and GAD-7 questionnaires, respectively. We hypothesised that there is an association between body weight and questionnaire scores, with increasing significance as BMI rises. Additionally, we hypothesised that the use of anti-obesity medication and bariatric surgery status would impact GAD-7 and PHQ-9 scores. 

As we anticipated, both GAD-7 and PHQ-9 scores demonstrated a positive correlation with BMI. This is consistent with prior studies using these assessment tools [28,29,30] and aligned with meta-analyses indicating a relationship between body weight and depressive and anxiety disorders [6,7]. This relationship is complex, with current understanding suggesting an interplay of psychological and biological pathways [6]. Obesity contributes to heightened psychological distress. Cultural norms surrounding beauty ideals may amplify body dissatisfaction and lower self-esteem, both of which are linked to depression and anxiety [7,31,32]. Women with obesity are more likely to be dissatisfied with their weight and experience social problems as compared to women of culturally acceptable weight [33]. Disrupted eating patterns, eating disorders, and physical discomfort stemming from obesity elevate depression risk and psychological distress [34,35]. Sleep disturbances are also significant, as sleep loss can contribute to the maintenance and/or exacerbation of anxiety and depression [36]. Notably, disruption of sleep associated with major depressive episodes is a significant factor in weight gain [37]. Looking at the pathophysiology, obesity can be understood as a chronic low-grade inflammation of fatty tissue, and activating inflammation pathways could lead to developing depression and anxiety [38,39,40]. Obesity often involves the dysregulation of the hypothalamic-pituitary-adrenal axis, a well-established factor in both of the disorders [41,42]. Its chronic activation is associated with chronically stressful or traumatic experiences, immunosuppression, and alteration in monoaminergic pathways, including noradrenaline, dopamine, and serotonin. Increased body weight also increases the risk of insulin resistance, potentially leading to brain alterations and heightened depression risk [43]. People with a larger waist circumference and more social anxiety symptoms had greater inflammation and insulin resistance relative to those with a larger waist circumference but less social anxiety symptoms [44]. Dysregulated mesolimbic dopamine signalling is implicated in both obesity and major depressive disorder (MDD), with individuals showing alterations in dopamine receptor availability and responsivity to rewards [45]. 

The scores also showed a negative correlation with age. This is in line with different studies using GAD-7 [46,47]. For PHQ-9, this may be due to the relatively young age of the participants. In a study on 15,847 participants, the average values of the PHQ-9 total scores followed a reverse U-shaped pattern: starting low during young adulthood, increasing during middle adulthood, and then decreasing during older adulthood [48]. Since the average age of our participants was 38.89 (SD = 9.00), we might have only observed a part of this trajectory. Similar correlations for both questionnaires might be due to the fact that there are a few similar items in both subscales regarding being restless or having difficulty sleeping or relaxing. In addition, both depression and anxiety symptoms are part of the same mood disorder category, share a common domain of negative affect, and share a cognitive process with negative bias in information processing. This finds confirmation in other research utilising these tools [49,50].

In our study, patients undergoing semaglutide treatment demonstrated reduced scores for both anxiety and depression symptoms. Semaglutide, on average, causes more significant weight reduction than liraglutide or bupropion with naltrexone [51]. This may be one of the reasons why its relationship with depression and anxiety symptoms is the strongest. Additionally, research suggests that changes in neurotransmitter expression are specific to GLP-1 agonist treatment. The distribution of semaglutide in the brain differed from that of liraglutide, particularly in the paraventricular nucleus of the hypothalamus and the lateral septal nucleus [52]. While evidence supporting the correlation between semaglutide and anxiety and depression symptoms is limited, there is more comprehensive data regarding liraglutide. In randomised controlled phase 2 and 3a trials with patients on a 3.0 mg dose, mean baseline Patient Health Questionnaire-9 scores of 2.8 ± 3.0 vs. 2.9 ± 3.1 for liraglutide vs. placebo improved to 1.8 ± 2.7 vs. 1.9 ± 2.7, respectively, at treatment end, supporting the neuropsychiatric safety of this class of medication [53]. In another study involving women with PCOS undergoing liraglutide 1.8 mg treatment, psychological health markers improved compared with the control group after the treatment [54]. It is worth noting that psychiatric adverse events from using incretin medications are generally rare and, when studied, comprised only 1.2% of the total reports for semaglutide, liraglutide, and tirzepatide [55]. However, there have been reports of semaglutide-associated depression, which resolved after withdrawing the medication [56]. Recently, there have been concerns about the impact of anti-obesity medication on mental health, specifically after patients prescribed semaglutide anecdotally reported suicidal ideations [57]. This has spurred European regulatory agencies to investigate this potential association. However, comprehensive studies have not supported these claims, as semaglutide was associated with a lower risk of suicidal thoughts than other anti-obesity and anti-diabetes drugs [58]. Nausea is one of the most common side effects of incretin medications and can possibly also contribute to mood disorders in treated individuals, as in other studies, severe nausea and vomiting were associated with a higher frequency of depression diagnosis, for example, in pregnant women [59]. On top of the benefits stemming from weight loss, bupropion is also an antidepressant medication acting as a norepinephrine and dopamine reuptake inhibitor that directly stimulates POMC cells [52]. However, in our study, there was no significant relationship between its use and PHQ-9 and GAD-7 scores. This might be due to the relatively small sample of patients using it, as its role in clinical settings has decreased following the development of new incretin medications.

Women who underwent bariatric surgery screened less frequently for anxiety in our study. A recent meta-analysis from 2022 also suggests improvement in mood symptoms after post-bariatric surgery [60]. The pooled proportion of patients with anxiety symptoms reduced from 24.5% pre-operatively to 16.9% post-operatively. There were significant reductions in the Generalised Anxiety Disorder Assessment-7 score of 0.54. The pooled proportion of depressive symptoms reduced from 34.7% pre-operatively to 20.4% post-operatively. There were also significant reductions in the Patient Health Questionnaire-9 score. The mean follow-up duration of post-bariatric surgery was 34 months. Several studies have reported less favourable long-term mental health outcomes post-bariatric surgery, with some stating that the initial improvement of depressive symptoms was not sustained beyond the first post-operative year [61]. The prevalence of post-bariatric surgery depression is also relatively high, with almost one in five patients affected by it. It could be dependent on the type of procedure. In one retrospective cohort study, admissions with anxiety disorders, any psychiatric diagnosis, and psychiatric inpatients increased after sleeve gastrectomy and gastric bypass but not after restrictive bariatric procedures (gastric banding/gastroplasty) [62]. Depression might be related to weight regain, eating disorders, and quality of life [63]. Additionally, it is suspected that long-term malabsorption might be related to the incidence of major depressive disorder after bariatric surgery, but the possible causal relationship between nutritional deficiency after bariatric surgery and major depressive disorder needs more investigation [64]. The evidence of moderate research shows that the risk of suicide and self-harm increases after bariatric surgery [65]. Thus, we are not drawing arbitrary conclusions from our observations and are convinced that this area requires further research.

The authors acknowledge the limitations of this study, including the lack of representativeness of the study group and the inability to reach individuals without internet access or those outside of themed support groups. Some studies have suggested that depression can influence the validity of self-reported BMI in the obese population and that women especially underestimate their BMI [66]. Moreover, this study was cross-sectional, and as such, no causal pathways could be investigated. We had no information on previous medical history, including the use of antidepressant medication. We were also unaware of the timeframe for the use of anti-obesity medication or bariatric surgery. Finally, it is important to note that while the PHQ-9 and GAD-7 are cost-effective screening instruments, they are not sufficient substitutes for a clinical diagnosis of depression and anxiety, respectively. Psychological support for participants was not provided.

## 5. Conclusions

Interdisciplinary assessment of patients living with obesity conducted by the therapeutic team, including mental health professionals, can address all factors leading to the development of obesity or influencing its treatment. A more profound investigation of semaglutide’s use is needed due to the promising evidence suggesting a positive effect on decreasing the severity of depression and anxiety symptoms to assess the direct or indirect character of this influence. The linkage between bariatric surgery and mental health status requires further research.

## Figures and Tables

**Table 1 nutrients-16-02438-t001:** Characteristics of the study group.

Variable		N/M ± SD
Age [years]		38.89 ± 9.00
Body mass index [kg/m^2^]		33.32 ± 6.67
Body mass index	Underweight	0 (0.0)
	Normal weight	92 (8.3)
	Overweight	254 (23.0)
	Obesity I	366 (33.1)
	Obesity II	247 (22.4)
	Obesity III	146 (13.2)
Pharmacological treatment of obesity		488 (44.2)
Pharmacological treatment of obesity	Semaglutide	232 (21.0)
	Liraglutide	235 (21.3)
	Naltrexone/Bupropione	41 (3.7)
Surgical treatment of obesity		131 (11.9)
Chronic diseases		327 (29.6)
Chronic diseases	Hypertension	220 (19.9)
	Cardiovascular diseases other than hypertension	57 (5.2)
	Obesity	496 (44.8)
	Hypothyroidism	387 (35.0)
	Diabetes mellitus type 2	116 (10.5)
	Osteoarthritis	76 (6.9)
	Depression	211 (19.1)
	Anxiety	153 (13.8)
	Dyslipidemia	101 (9.1)
	Fatty liver disease	112 (10.1)
	Other	367 (33.2)

M—mean, SD—Standard deviation, N—number, kg/m^2^—kilogram (s) per square metre.

**Table 2 nutrients-16-02438-t002:** Summary of results and interpretation of the Generalised Anxiety Disorder-7 and Patient Health Questionnaire-9.

Variable		N/M ± SD
GAD-7		9.57 ± 6.10
GAD-7 Interpretation	Anxiety	513 (46.4)
PHQ-9		10.6 ± 6.36
PHQ-9 Interpretation	Depression	556 (50.3)

M—mean, SD—Standard deviation, N—number, Questionnaire, GAD-7—Generalised anxiety disorder, PHQ-9—Patient Health Questionnaire-9.

**Table 3 nutrients-16-02438-t003:** Correlations between PHQ-9 and GAD-7 scores with age and BMI.

Variable	M ± SD/r	*p* *
GAD-7		
Age [years]	−0.106	**<0.001**
BMI [kg/m^2^]	0.121	**<0.001**
PHQ-9		
Age [years]	−0.103	**<0.001**
BMI [kg/m^2^]	0.173	**<0.001**

M—mean, SD—Standard deviation, N—number, BMI—body mass index, kg/m^2^—kilogram (s) per square meter, PHQ-9—Patient Health Questionnaire-9, * Spearman’s rank correlation, Significant effects (<0.05) are marked in bold.

**Table 4 nutrients-16-02438-t004:** Overview of GAD-7 scores.

Variable		GAD-7				
		M ± SD/r	*p*	Anxiety N(%)	No Anxiety N(%)	*p*
BMI	Normal weight	8.41 ± 5.48	**<0.001%**	32 (34.8)	60 (65.2)	**<0.** **001 &**
	Overweight	9.22 ± 6.09		110 (43.3)	144 (56.7)	
	Obesity I	8.86 ± 6.15		152 (41.5)	214 (58.5)	
	Obesity II	10.69 ± 6.09		136 (55.1)	111 (44.9)	
	Obesity III	10.79 ± 5.97		83 (56.9)	63 (43.1)	
Pharmacological treatment of obesity	Yes	9.36 ± 6.01	0.321 *	221 (45.3)	267 (54.7)	0.499 &
	No	9.74 ± 6.17		292 (47.3)	325 (52.7)	
Semaglutide	Yes	8.71 ± 6.16	**0.013 ***	100 (43.1)	132 (56.9)	0.254 &
	No	9.80 ± 6.07		413 (47.3)	460 (52.7)	
Liraglutide	Yes	9.78 ± 5.85	0.465 *	110 (46.8)	125 (53.2)	0.8944 &
	No	9.52 ± 6.17		403 (46.3)	467 (53.7)	
Naltrexone/Bupropione	Yes	10.54 ± 6.45	0.365 *	20 (48.8)	21 (51.2)	0.757 &
	No	9.55 ± 6.10		493 (46.3)	571 (53.7)	
Surgical treatment of obesity	Yes	8.03 ± 6.27	**0.002 ***	47 (35.9)	84 (64.1)	**0.009 &**
	No	9.78 ± 6.06		466 (47.8)	508 (52.2)	

M—mean, SD—Standard deviation, N—number, BMI—body mass index, kg/m^2^—kilogram (s) per square meter, PHQ-9—Patient Health Questionnaire-9, * Spearman’s rank correlation, & Chi square test, Significant effects (<0.05) are marked in bold.

**Table 5 nutrients-16-02438-t005:** Overview of PHQ-9 scores.

Variable		PHQ-9				
		M ± SD/r	*p*	Depression N(%)	No Depression N(%)	*p*
BMI	Normal weight	8.67 ± 6.26	**<0.001%**	30 (32.6)	62 (67.4)	**<0.001 &**
	Overweight	9.97 ± 6.23		111 (43.7)	143 (56.3)	
	Obesity I	10.05 ± 6.15		168 (45.9)	198 (54.1)	
	Obesity II	11.89 ± 6.36		153 (61.9)	94 (38.1)	
	Obesity III	12.22 ± 6.46		94 (64.4)	52 (35.6)	
Pharmacological treatment of obesity	Yes	10.39 ± 6.02	0.329 *	229 (46.9)	259 (53.1)	**0.044 &**
	No	10.79 ± 6.22		327 (53.0)	290 (47.0)	
Semaglutide	Yes	9.76 ± 6.37	**0.013 ***	102 (43.9)	130 (56.1)	**0.029 &**
	No	10.84 ± 6.33		454 (52.0)	419 (48.0)	
Liraglutide	Yes	10.92 ± 5.98	0.253 *	116 (49.4)	119 (50.6)	0.741 &
	No	10.52 ± 6.45		440 (50.6)	430 (49.4)	
Naltrexone/Bupropione	Yes	11.09 ± 6.32	0.683 *	21 (51.2)	20 (48.8)	0.906 &
	No	10.59 ± 6.36		535 (50.3)	529 (49.7)	
Surgical treatment of obesity	Yes	9.77 ± 6.29	0.109 *	61 (46.6)	70 (53.4)	0.361 &
	No	10.73 ± 6.36		495 (50.8)	479 (49.2)	

M—mean, SD—Standard deviation, N—number, BMI—body mass index, kg/m^2^—kilogram (s) per square meter, PHQ-9—Patient Health Questionnaire-9, * Spearman’s rank correlation, & Chi square test, Significant effects (<0.05) are marked in bold.

## Data Availability

The data presented in this study are available on request from the corresponding author.

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
