# Peer review of "Association between Symptoms of Depression and Generalised Anxiety Disorder Evaluated through PHQ-9 and GAD-7 and Anti-Obesity Treatment in Polish Adult Women"

_nutrients, 2024, doi:10.3390/nu16152438_

Round 1

Reviewer 1 Report

Comments and Suggestions for Authors

The manuscript presents results of an interesting study on the effect of obesity on the depression and generalized anxiety disorder. The study was done on a fairly large population of 1105 female participants. The study was well designed. The results are properly, professionally described and discussed.

The reasoning and Conclusions are scientifically sound.

Remarks

I wonder whether  this manuscript fits the profile of the journal. It is submitted to the Special Issue on Bidirectional Link between Eating Habits, Lifestyle, Physical Exercise and Depression and Other Mental Disorders but the question of eating habits appears in the questionnaire but is not addressed in the Results. 

Materials and Methods contain description of the inclusion criteria. Were there any exclusion criteria? The authors state that incomplete responses were excluded from the analysis; it was an exclusion criterion so perhaps the initial number of participants was higher than the final numer reported.

It would be of interest if the level of education and environment (big city, small town, village) affected the results but these factors were not include in the study so this remark may be only a suggestion for future studies.

The authors analyzed the effect of obesity treatment; have they also analyze the effects of accompanying diseases?

What were the BMI criteria for classification of participants into respective groups? Apparently standard, but they could be recalled.

Author Response

Dear Reviewer,

Thank you for taking your time to review our article and for your valuable feedback. We have carefully considered all your comments and made following revisions:

Comment 1: “I wonder whether  this manuscript fits the profile of the journal. It is submitted to the Special Issue on Bidirectional Link between Eating Habits, Lifestyle, Physical Exercise and Depression and Other Mental Disorders but the question of eating habits appears in the questionnaire but is not addressed in the Results.”

Response 1: In our manuscript, we have focused on the relationship between depression, anxiety, obesity and its treatment and their bidirectional character and complex interplay, which we believe fits the profile of this special issue. We have explored the subject of eating habits in a different paper, utilizing the TFEQ-18 scale, which provided additional insight. We understand your concerns, but we also trust that this manuscript would be an interesting addition as it also provides valuable clinical context. 

Comment 2: “Materials and Methods contain description of the inclusion criteria. Were there any exclusion criteria? The authors state that incomplete responses were excluded from the analysis; it was an exclusion criterion so perhaps the initial number of participants was higher than the final numer reported.”

Response 2: We have added additional explanation in the methods section: “The exclusion criteria were being male, not giving consent and providing incompetent responses in the questionnaire“. This can be found on page 3, lines 115-117.

Comment 3: “It would be of interest if the level of education and environment (big city, small town, village) affected the results but these factors were not include in the study so this remark may be only a suggestion for future studies.”

Response 3: As the data collection has finished, we are unable to expand the questionnaire. However, we agree that this would be a very interesting addition to the study and will try to implement it in our future research.

Comment 4: “The authors analyzed the effect of obesity treatment; have they also analyzed the effects of accompanying diseases?”

Response 4: In this manuscript, we primarily focused on obesity and its relationship with mental disorders. We agree that comorbidities could be potential confounding factors, and it is definitely worth exploring this in our future work. 

Comment 5: “What were the BMI criteria for classification of participants into respective groups? Apparently standard, but they could be recalled.”

Response 5: We have added additional explanation to the methods section: ““The participants were divided into six groups based on their calculated Body Mass Index (BMI): underweight (BMI below 18.5), normal weight (BMI 18.5 to 24.9), overweight (BMI 25.0 to 29.9), Obesity I (BMI 30.0 to 34.9), Obesity II (BMI 35.0 to 39.9), and Obesity III (BMI above 40)”. This can be found on page 3, lines 117-120.

We believe that these additions  have significantly improved the manuscript and we appreciate the constructive nature of your feedback. You will find the revised manuscript attached.

Thank you for your consideration.

Sincerely,

Tomasz Witaszek

Reviewer 2 Report

Comments and Suggestions for Authors

The study is interesting, but it can be improved with some inclusions 

Abstract - Looks fine 

Introduction: "WHO report from 2022, almost 60% of citizens in Europe are either over-weight or obese"  - please elaborate beyond the Europe too 

"increasing focus on the relationship between body weight and mental health disorders, such as depression and anxiety" : what's the reasons, social media, excessive body focus, body dysmorphophobia, need to look better, social modelling, psychological sick role, over interpretation of possible healthy weight. 

You mentioned biological cause of Depression in Obesity but some obese do not develop due to cultural reasons, then how strongly you rely on biological issues > need to develop a debate on it. 

Association between Anxiety and Obesity : less investigated, is it because anxiety is invariably present in Depressive disorders 

what about social anxiety 

Social Anxiety Symptoms Moderate the Link Between Obesity and Metabolic Function - PMC (nih.gov)

OCD 

Body mass index in obsessive-compulsive disorder - ScienceDirect

Fat phobia scale-short form and beliefs about obese persons scale: cross-cultural adaptation to Brazilian Portuguese | Discover Psychology (springer.com)

Obesity OSA and Anxiety 

Factors predicting the presence of depression in obstructive sleep apnea - PubMed (nih.gov)

gender anxiety 

Body Mass Index Distributions and Obesity Prevalence in a Transgender Youth Cohort – A Retrospective Analysis - ScienceDirect

PHQ 9 was used 

Why author did not use BDI self-rating scale 

Antti obesity medicines - any biological mechanism to protect from depression 

Liraglutide VS Semaglutide : why the outcomes are different 

Can author do regression analysis to check if there is a factor which reduces anxiety / depression 

Can author do an analysis by log transformation and parametric analysis then compare with non-parametric data outcome 

Surgical treatment of obesity > there are research where it's seen that it worsens, any typology of surgery, Incidence of adverse mental health outcomes after sleeve gastrectomy compared with gastric bypass and restrictive bariatric procedures: a retrospective cohort study - Sumithran - 2023 - Obesity - Wiley Online Library   

In light of those, how will you formulate your discussion 

Does the side effects like significant nausea from Glutide can affect mental health 

Author Response

Dear Reviewer,

Thank you for taking your time to review our article and for your valuable feedback. We have carefully considered all your comments and made following revisions:

Comment 1: “Introduction: "WHO report from 2022, almost 60% of citizens in Europe are either overweight or obese"  - please elaborate beyond Europe too”.

Response 1: We made changes to the introduction to provide a more global perspective. It now states that  “Obesity, a multifactorial and complex disease, has reached pandemic proportions. According to the WHO report from 2022, 1 in 8 people in the world were living with obesity. Its prevalence varies by region, from 31% in the South-East Asia Region and the African Region to 67% in the Region of the Americas [1].” This change can be found on page 1, lines 30-33.

Comment 2: “Increasing focus on the relationship between body weight and mental health disorders, such as depression and anxiety" : what's the reasons, social media, excessive body focus, body dysmorphophobia, need to look better, social modeling, psychological sick role, over interpretation of possible healthy weight”.

Response 2: We have made an addition in the introduction to cover this relationship. It now states that “In recent years, there has been an increasing focus on the relationship between body weight and mental health disorders, driven by their rising prevalence, socioeconomic factors, and heightened public awareness (including the influence of social media, excessive focus on body image, body dysmorphophobia, and misinterpretation of healthy weight standards) [5-9].” This change can be found on page 1, lines 39-43.

Comment 3: “You mentioned the biological cause of Depression in Obesity but some obese do not develop due to cultural reasons, then how strongly you rely on biological issues > need to develop a debate on it.”

Response 3: We have mentioned cultural impact in our introduction which covers this subject: “The observed variations in results across different continents can suggest that the relationship between body size and depression may be influenced by cultural conception of the ideal body weight and related social pressures on the relationship between body size and depression. In cultures where a larger body size is considered the norm, obesity might be more socially acceptable, leading to lower levels of body dissatisfaction, reduced mental stress, and acting as a buffer against weight discrimination [7,13].“ This can be found on page 2, lines 62-68.

Comment 4: “Association between Anxiety and Obesity : less investigated, is it because anxiety is invariably present in Depressive disorders”

Response 4: Historically, depression  has long been recognized as a major public health issue associated with obesity, driving more research attention and funding toward this area. Anxiety as disease is more complex and this diversity can make it challenging to study this relationship. We expressed that by changing our introduction: “The association between body weight and anxiety disorder is less studied, which can be due to its less direct and measurable impacts on mortality and morbidity. “ This can be found on page 2, lines 69-70.

Comment 5: “What about social anxiety - Social Anxiety Symptoms Moderate the Link Between Obesity and Metabolic Function - PMC (nih.gov)”.

Response 5: We have added the following citation to provide social anxiety context: “People with a larger waist circumference and more social anxiety symptoms had greater inflammation and insulin resistance relative to those with a larger waist circumference but less social anxiety symptoms [44].” This can be found on page 7, lines 221 - 223.

Comment 6: “OCD - Body mass index in obsessive-compulsive disorder - ScienceDirecT, Fat phobia scale-short form and beliefs about obese persons scale: cross-cultural adaptation to Brazilian Portuguese | Discover Psychology (springer.com)”.

Response 6: To expand our introduction by elaborating on OCD, we have added: “Interestingly, OCD is associated with significantly lower rates of obesity and overweight, but this relationship was not found when comorbid depression was present [18].” This can be found on page 2, lines 78-80.

Comment 7: “Obesity OSA and Anxiety - Factors predicting the presence of depression in obstructive sleep apnea - PubMed (nih.gov)”

Response 7:  Relationship between OSA and Anxiety is worth exploring, therefore we have added: “The co-occurrence of depression and obesity can also complicate the recognition and treatment of its complications, such as sleep apnea [12].” This is page 2, lines 61-62.

Comment 8: “Gender anxiety - Body Mass Index Distributions and Obesity Prevalence in a Transgender Youth Cohort – A Retrospective Analysis - ScienceDirect”

Response 8: To elaborate on transgender population,  we have added: “There were significantly higher rates of overweight and obesity in transgender population, with the results being more pronounced in those assigned female at birth [11]” This change can be seen on page 2, lines 59-61.

Comment 9: “PHQ 9 was used  - Why author did not use BDI self-rating scale”

Response 9: We have decided on using PHQ-9 as it has satisfactory psychometric properties and demonstrated a similar ability to monitor changes in the severity of depression during treatment. The authors of the study “Ocena psychometrycznych wÅ‚aÅ›ciwoÅ›ci polskiej wersji Kwestionariusza Zdrowia Pacjenta-9 dla osób dorosÅ‚ych” ( Kokoszka, A., JastrzÄ™bski, A., & ObrÄ™bski, M. (2016). Psychiatria, 13(4), 187–193. https://journals.viamedica.pl/psychiatria/article/view/49966/38212) highlighted the advantage of the PHQ-9 over the BDI-II in its significantly shorter form and its reference to official diagnostic criteria. Better psychometric properties of the PHQ-9 compared to HADS and WHO-5 have been demonstrated."

Comment 10: “Anti obesity medicines - any biological mechanism to protect from depression”.

Response 10: We have expanded this topic in the discussion, adding: “On top of the benefits stemming from weight loss, bupropion is also an antidepressant medication acting as norepinephrine and dopamine reuptake inhibitor that directly stimulate POMC cells [52]. However, in our study, there was no significant relationship between its use and PHQ-9 and GAD-7 scores. This might be due to the relatively small sample of patients using it, as its role in clinical settings has decreased following the development of new incretin medications.” This can be found on page 8, lines 267-272.

Comment 11: “Liraglutide VS Semaglutide : why the outcomes are different” 

Response 11: We have added to this topic in the discussion: “ Semaglutide, on average, causes more significant weight reduction than liraglutide or bupropion with naltrexone [51]. This may be one of the reasons why its relationship with depression and anxiety symptoms is the strongest. Additionally, research suggests that changes in neurotransmitter expression are specific to GLP-1 agonist treatment. The distribution of semaglutide in the brain differed from that of liraglutide, particularly in the paraventricular nucleus of the hypothalamus and the lateral septal nucleus [52].” page 7, lines 240-246

Comment 12: “Surgical treatment of obesity > there are research where it's seen that it worsens, any typology of surgery, Incidence of adverse mental health outcomes after sleeve gastrectomy compared with gastric bypass and restrictive bariatric procedures: a retrospective cohort study - Sumithran - 2023 - Obesity - Wiley Online Library”.

Response 12: We have expanded this topic in the discussion, by adding: “The prevalence of post-bariatric surgery depression is also relatively high with almost one in five patients affected by it. It could be dependent on the type of procedure. In one retrospective cohort study, admissions with anxiety disorders, any psychiatric diagnosis, and as psychiatric inpatients increased after sleeve gastrectomy and gastric bypass, but not after restrictive bariatric procedures (gastric banding/gastroplasty) [62]. “ This can be found on page 8, lines 283-288

Comment 13: “Does the side effects like significant nausea from Glutide can affect mental health”

Response 13: We have elaborated this in the discussion and added: “Nausea is one of the most common side effects of incretin medications and can possibly also contribute to mood disorders in treated individuals, as in other studies severe nausea and vomiting were associated with a higher frequency of depression diagnosis, for example in pregnant women [59].” This can be found on page 8, lines 263-267.

We believe that these additions  have significantly improved the manuscript and we appreciate the constructive nature of your feedback. You will find the revised manuscript attached.

Thank you for your consideration.

Sincerely,

Tomasz Witaszek

Reviewer 3 Report

Comments and Suggestions for Authors

Dear Authors,

I would like to thank you for the article titled "Association between Symptoms of Depression and Generalised Anxiety Disorder Evaluated through PHQ-9 and GAD-7 and Anti-obesity Treatment in Polish Adult Women". Your work represents a significant contribution to understanding the complex interactions between obesity, depression, and anxiety. Your efforts in data collection and analysis are appreciated, and your study adds value to the existing literature. The detailed information and in-depth analysis you have provided are extremely useful for health professionals seeking to address obesity from both a physical and psychological perspective.

Strengths of the Article:

  1. Interdisciplinary Approach: The inclusion of mental health professionals in the therapeutic team highlights a holistic approach to obesity treatment, considering both the physical and psychological aspects of the disease.
  2. Use of Validated Tools: Utilizing the PHQ-9 and GAD-7 scales, validated tools for diagnosing and assessing depression and anxiety, adds credibility to the results, ensuring that the measurements are standardized and reliable.
  3. Relevant Study Sample: The large sample of 1,105 adult women provides a solid basis for statistical analysis and generalization of the results.
  4. Detailed Analysis: The correlation between BMI and anxiety and depression scores is explored in detail, offering deeper insights into the dynamics between obesity and mental health.
  5. Clinical Implications: The study underscores the importance of considering psychological factors in obesity treatment, suggesting potential improvements in clinical practices.

Lines to Modify and How to Modify Them:

  1. Abstract (p. 1, line 11):

    • Line to modify: "The sample consisted of 1,105 adult women."
    • Suggested modification: "The study sample comprised 1,105 adult women."
  2. Introduction (p. 1, line 1):

    • Line to modify: "Obesity, a complex and multifactorial disease, has reached pandemic dimensions."
    • Suggested modification: "Obesity, a multifaceted and complex disease, has reached pandemic proportions."
  3. Introduction (p. 2, line 1):

    • Line to modify: "The association between body weight and anxiety disorder is less investigated, however a recent meta-analysis reveals that individuals with obesity and overweight have more anxiety compared to people without obesity, and that the relationship between them is stronger in women than in men."
    • Suggested modification: "The association between body weight and anxiety disorder is less studied; however, a recent meta-analysis reveals that individuals with obesity and overweight have higher levels of anxiety compared to those without obesity, and that this relationship is stronger in women than in men."
  4. Methods (p. 2, line 15):

    • Line to modify: "The inclusion criteria were being female, aged 18 or older, residing in Poland, and having internet access."
    • Suggested modification: "Inclusion criteria included being female, aged 18 or older, residing in Poland, and having internet access."
  5. Results (p. 4, line 5):

    • Line to modify: "Among the population, 29.6% had chronic diseases, with obesity being the most common (44.8%, n = 496)."
    • Suggested modification: "Within the study population, 29.6% had chronic diseases, with obesity being the most prevalent (44.8%, n = 496)."
  6. Discussion (p. 6, line 1):

    • Line to modify: "This connection is complex, with current understanding suggesting an interplay between psychological and biological pathways."
    • Suggested modification: "This relationship is complex, with current understanding suggesting an interplay of psychological and biological pathways."
  7. Discussion (p. 6, line 10):

    • Line to modify: "The authors are aware of the limitations of this study, starting with the lack of representativeness of the study group and inability to reach individuals without internet access or outside of themed support groups."
    • Suggested modification: "The authors acknowledge the limitations of this study, including the lack of representativeness of the study group and the inability to reach individuals without internet access or those outside of themed support groups."

Parts to Eliminate:

  1. Methods (p. 2, line 20):

    • Line to eliminate: "Incomplete responses were excluded from the analysis."
    • Reason: This detail is implicit in most studies and is redundant.
  2. Discussion (p. 6, line 20):

    • Line to eliminate: "It's important to underline that our study didn’t specify when the surgery took place."
    • Reason: This statement is repetitive of the limitations already discussed.
    • I would like to suggest that you include the following citation in your article to reinforce the validity of the psychometric tools used:

      Cavicchiolo, E., Sibilio, M., Lucidi, F., Cozzolino, M., Chirico, A., Girelli, L., Manganelli, S., Giancamilli, F., Galli, F., Diotaiuti, P., Zelli, A., Mallia, L., Palombi, T., Fegatelli, D., Albarello, F., & Alivernini, F. (2022). The Psychometric Properties of the Behavioural Regulation in Exercise Questionnaire (BREQ-3): Factorial Structure, Invariance and Validity in the Italian Context. International journal of environmental research and public health, 19(4), 1937. https://doi.org/10.3390/ijerph19041937

      Instructions on How and Where to Include the Citation

      Section to Modify: Methods

      Page: 2

      Position: After the description of the PHQ-9 and GAD-7 scales

      Original Text:2.3. GAD-7 scale

      The 7-item Generalised Anxiety Disorder Scale (GAD-7) was developed as a screening tool designed to identify generalised anxiety disorder (GAD), particularly in primary care settings [20]. It assesses the frequency of experiencing seven distinct symptoms of anxiety over the past two weeks. Response options include "not at all," "several days," "more than half the days," and "nearly daily," scored as 0, 1, 2, and 3, respectively [20]. The Polish translation of the GAD-7, provided by the MAPI Research Institute, was used [21]. Internal consistency, measured by Cronbach's alpha, was found to be high at 0.924, indicating strong reliability.

      Modified Text:

      2.3. GAD-7 scale

      The 7-item Generalised Anxiety Disorder Scale (GAD-7) was developed as a screening tool designed to identify generalised anxiety disorder (GAD), particularly in primary care settings [20]. It assesses the frequency of experiencing seven distinct symptoms of anxiety over the past two weeks. Response options include "not at all," "several days," "more than half the days," and "nearly daily," scored as 0, 1, 2, and 3, respectively [20]. The Polish translation of the GAD-7, provided by the MAPI Research Institute, was used [21]. Internal consistency, measured by Cronbach's alpha, was found to be high at 0.924, indicating strong reliability. It is crucial to ensure the psychometric robustness of these tools, as demonstrated in various contexts [Cavicchiolo et al., 2022]

Author Response

Dear Reviewer,

Thank you for taking the time to review our article and for your valuable feedback. We have carefully considered all of your comments and agree with them. We have implemented all suggested revisions in our article.

We believe that these additions  have significantly improved the manuscript and we appreciate the constructive nature of your feedback. You will find the revised manuscript attached.

Thank you for your consideration.

Sincerely,

Tomasz Witaszek